# The Pluripotency Factor Nanog Protects against Neuronal Amyloid β-Induced Toxicity and Oxidative Stress through Insulin Sensitivity Restoration

**DOI:** 10.3390/cells9061339

**Published:** 2020-05-27

**Authors:** Ching-Chi Chang, Hsin-Hua Li, Sing-Hua Tsou, Hui-Chih Hung, Guang-Yaw Liu, Tatiana A. Korolenko, Te-Jen Lai, Ying-Jui Ho, Chih-Li Lin

**Affiliations:** 1Institute of Medicine, Chung Shan Medical University, Taichung 402367, Taiwan; fmaj7@seed.net.tw (C.-C.C.); vivid529@hotmail.com (H.-H.L.); zinminid@gmail.com (S.-H.T.); liugy@csmu.edu.tw (G.-Y.L.); tejenlai@hotmail.com (T.-J.L.); 2Department of Psychiatry, Chung Shan Medical University Hospital, Taichung 402367, Taiwan; 3Department of Medical Research, Chung Shan Medical University Hospital, Taichung 402367, Taiwan; 4Department of Life Sciences and Institute of Genomics and Bioinformatics, National Chung Hsing University, Taichung 402204, Taiwan; hchung@dragon.nchu.edu.tw; 5Scientific Research Institute of Physiology and Basic Medicine, Novosibirsk 630117, Russia; t.a.korolenko@physiol.ru; 6Department of Psychology, Chung Shan Medical University, Taichung 402367, Taiwan

**Keywords:** amyloid β, insulin signaling, oxidative stress, Nanog, senescence

## Abstract

Amyloid β (Aβ) is a peptide fragment of the amyloid precursor protein that triggers the progression of Alzheimer’s Disease (AD). It is believed that Aβ contributes to neurodegeneration in several ways, including mitochondria dysfunction, oxidative stress and brain insulin resistance. Therefore, protecting neurons from Aβ-induced neurotoxicity is an effective strategy for attenuating AD pathogenesis. Recently, applications of stem cell-based therapies have demonstrated the ability to reduce the progression and outcome of neurodegenerative diseases. Particularly, Nanog is recognized as a stem cell-related pluripotency factor that enhances self-renewing capacities and helps reduce the senescent phenotypes of aged neuronal cells. However, whether the upregulation of Nanog can be an effective approach to alleviate Aβ-induced neurotoxicity and senescence is not yet understood. In the present study, we transiently overexpressed Nanog—both in vitro and in vivo—and investigated the protective effects and underlying mechanisms against Aβ. We found that overexpression of Nanog is responsible for attenuating Aβ-triggered neuronal insulin resistance, which restores cell survival through reducing intracellular mitochondrial superoxide accumulation and cellular senescence. In addition, upregulation of Nanog expression appears to increase secretion of neurotrophic factors through activation of the Nrf2 antioxidant defense pathway. Furthermore, improvement of memory and learning were also observed in rat model of Aβ neurotoxicity mediated by upregulation of Nanog in the brain. Taken together, our study suggests a potential role for Nanog in attenuating the neurotoxic effects of Aβ, which in turn, suggests that strategies to enhance Nanog expression may be used as a novel intervention for reducing Aβ neurotoxicity in the AD brain.

## 1. Introduction

Alzheimer’s disease (AD) is a common neurodegenerative disease accounting for half of all dementia cases. Although the pathogenesis of AD is quite complex and not fully understood, the increase of extracellular senile plaques in the brain acts as a central hallmark of disease progression. It is known that the major protein component of the senile plaques is a short peptide called amyloid β (Aβ) which causes neuronal cell death and thus leading to dementia [1]. Although several toxic pathways were implicated, it is known that Aβ plays a causative role of mitochondrial dysfunction and oxidative stress that in turn leads to neuronal dysfunction and apoptosis [2]. The deposition of Aβ is a significant pathologic feature seen in the aging brain [3], indicating Aβ may accelerate neuronal senescence [4]. In particular, the dysregulation of mitochondrial turnover is involved in brain senescence, demonstrating that impairment of mitochondrial function and increased oxidative damage also play an important role in AD-related pathology [5]. Actually, there is strong evidence that the greatest risk factor for AD is the increasing age [6]. This indicates therapeutic strategies to reduce Aβ toxicity can focus on slowing down or reversing the effects of neuronal aging [7]. Emerging evidence suggests that neuron aging is affected by the cell microenvironment and elicited by Aβ accumulation, which was shown to induce cellular senescence via generation of excessive levels of reactive oxygen species (ROS) [8]. In contrast, the potential of stem cell-based approaches in slowing down the aging process is relevant for neuroprotection in AD [9]. For, example, stem cells were demonstrated to serve as a protective source for neurotrophic factors secretion in attenuating Aβ-induced cytotoxicity and apoptosis [10]. Therefore, safeguarding the survival of neuronal cells by enhancing specific stemness gene expression may play a role in reducing toxic impact of Aβ in the brain.

It is known that type 2 diabetes (T2D) is another important non-genetic risk factor of AD, suggesting that alterations in insulin signaling may also be involved in AD pathogenesis [11]. In fact, recent studies have confirmed that insulin signaling is indeed impaired in the AD brain [12]. Particularly, impairments in brain insulin signaling transduction is associated with Aβ accumulation [13], indicating Aβ may contribute to brain insulin resistance. Previously, we have demonstrated that a pluripotency-associated miRNA miR-302 plays a neuroprotective role against Aβ toxicity [14]. We reported that the upregulation of miR-302 can suppress Aβ-induced insulin signaling blockade and maintain the intracellular redox balance toward cell survival. In addition, miR-302 also alleviated Aβ-mediated mitochondrial dysfunction thus diminishing oxidative and senescence-associated damages in neuronal cells. Interestingly, our data also revealed that genetic or pharmacologic stimulation of miR-302 upregulates high Nanog expression. Since Nanog is known as a transcriptional factor that enhances self-renewal, pluripotency [15] and antiaging [16] in stem cell biology, it is likely that Nanog itself may contribute to protect neuronal cells against Aβ toxicity. However, whether the upregulation of Nanog can be an effective way to reduce Aβ neurotoxicity is not yet understood. Therefore, in this study we investigated the protective mechanisms of Nanog. Our results showed that overexpression of Nanog is responsible for attenuating Aβ-triggered neuronal insulin resistance, which promotes cell survival through reducing intracellular ROS production and cellular senescence. This indicates upregulation of Nanog may promote survival of neuronal cells by attenuating Aβ neurotoxicity.

## 2. Materials and Methods

### 2.1. Materials

The human wild-type Nanog (#28221) expression vector was acquired from Addgene (Cambridge, MA, USA). All chemicals were purchased from Sigma (München, Germany). Primary antibodies used in this study: β-actin (NB600-501), Akt (sc-8312), p-Akt (sc-33437), GSK3β (sc-8257), p-GSK3β (sc-11757), Nrf2 (sc-722), caspase 3 (sc-7148) and poly (ADP-ribose) polymerase (PARP) (sc-7150) were obtained from Santa Cruz Biotechnology (Santa Cruz, CA, USA); SOD1 (#2770S) and Sirt1 (GTX61042) were purchased from GeneTex (Irvine, CA, USA); tau (05-348), p-tau (MAB5450) and p-Tyr (05-321) were obtained from Millipore (Bedford, MA, USA); Nanog (#4903), IRS-1 (194320) and p-IRS-1 (05–1087) were purchased from Cell Signaling Technology (Danvers, MA, USA). Aβ_1-42_ was synthesized by LifeTein (Somerset, NJ, USA) and soluble Aβ oligomers were prepared according to our previously described [17]. Briefly, Aβ was first dissolved in hexafluoroisopropanol (HFIP), distributed in aliquots, dried and stored at −80 °C. The day before the experiment, Aβ was dissolved in sterile dimethyl sulfoxide (DMSO), then diluted with PBS (pH 7.4) at a final concentration of 100 μM, vortexed and incubated at 4 °C for 24 h.

### 2.2. Cell Culture, Transfection and Viability Assay

SK-N-MC cells (human neuroblastoma cell line) were obtained from the American Type Culture Collection (Bethesda, MD, USA). Cells were cultured in Minimal Eagle’s Medium (MEM; Gibco) supplemented with 10% fetal calf serum, antibiotics (100 units/mL penicillin, 100 µg/mL streptomycin), 2-mM l-glutamine and kept in a humidified air containing 5% CO_2_ at 37 °C. Transient transfections were carried out using Lipofectamine 3000 reagent (Thermo Fisher Scientific, Waltham, MA, USA), according to manufacturer’s instruction. For viability assays, cells were treated with tetrazolium salt methyl-thiazol-tetrazolium (MTT) for 30 min, and then analyzed spectrophotometrically at 550 nm. Viability was determined as percent of control cells treated with vehicle alone.

### 2.3. mRNA Expression Analysis by Reverse-Transcription Quantitative PCR (qPCR)

Total mRNA was extracted from cultured cells or rat brain tissues using RNeasy Kit (Qiagen, Germantown, MD, USA). mRNAs were reverse-transcribed into cDNAs using TProfessional Thermocycler Biometra (Göttingen, Germany) following the manufacturer’s recommendations. qPCR was performed on an ABI 7300 Sequence Detection System (Applied Biosystems, Foster City, CA, USA), using Power SYBR Green PCR Master Mix (Applied Biosystems). The following temperature parameters were used: initial denaturation at 95 °C for 10 min, 40 cycles of denaturation at 95 °C for 15 s, annealing at 60 °C for 1 min and dissociation stage at 95 °C for 15 s, 60 °C for 15 s and 95 °C for 15 s. The following primer pairs were used: forward 5′-GACGT GTGAA GATGA GTGAA ACTGA-3′ and reverse 5′-GTTTC CAAAC AAGAA AAATC CTATG AG-3′ for Nanog, forward 5′-GCTGG CGATT CATAA GGATA GAC-3′ and reverse 5′-TATAC AACAT AAATC CACTA TCTTC CCCT-3′ for brain-derived neurotrophic factor (BDNF), forward 5′-TGCTT CCGGA GCTGT GATCT-3′ and reverse 5′-CGGAC AGAGC GAGCT GACTT-3′ for insulin-like growth factor 1 (IGF-1). The mRNA levels were normalized to glyceraldehyde 3-phosphate dehydrogenase (GAPDH) using primer pairs with forward 5′-TGGTAT CGTGG AAGGA CTCAT GAC-3′ and reverse 5′-ATGCC AGTGA GCTTC CCGTT CAGC-3′. Each cDNA sample was tested in triplicate. Values of relative mRNA expression were obtained by using Sequence Detection Systems software (Sequence Detection Systems 1.2.3-7300 Real-Time PCR System; Applied Biosystems) with a cycle threshold (delta-delta Ct) method.

### 2.4. Western Blot Analysis

Whole extracts of cells and rat brain tissues were prepared using Gold Lysis buffer (50-mM Tris-HCl, pH 8.0, 5-mM ethylenediaminetetraacetic acid, 150-mM NaCl, 0.5% Nonidet *P-*40, 0.5-mM phenylmethylsulfonyl fluoride and 0.5-mM dithiothreitol). Equal protein amounts from whole lysates were separated by 8–10% sodium dodecyl sulfate (SDS)-polyacrylamide gel electrophoresis and then transferred to polyvinylidene difluoride membranes (Millipore). The membranes were probed with primary antibodies followed by secondary antibodies conjugated with horseradish peroxidase. Images were acquired using Amersham ECL reagents and AI600 imager system (GE Healthcare, Chicago, IL, USA).

### 2.5. DAPI Staining on Nuclei

Cells were fixed in ice-cold methanol with 4% paraformaldehyde for 24 h. After that cells were incubated with 4’,6-diamidino-2-phenylindole (DAPI) solution (1 ng/mL in McIlvaine’s buffer, pH 7.0) for 15 min at room temperature and observed with a fluorescent microscope (DP80/BX53, Olympus, Tokyo, Japan) for apoptotic fragmented nuclei.

### 2.6. Detection of Superoxide by Dihydroethidium (DHE) Staining

DHE is a cell-permeable fluorogenic probe that reacts with superoxide to form ethidium and emits red fluorescence. At the end of experiments, cells were rinsed with PBS solution and then incubated in fresh media containing 10-μM DHE solution for 30 min in the dark at room temperature. After incubation, cells were washed twice with PBS and observed by using an inverted fluorescence microscope (DP72/CKX41, Olympus). All images were used the same fluorescent conditions and exposure time.

### 2.7. Analysis of Mitochondrial Membrane Potential by JC-1 Staining

Cells were incubated with fresh media containing 1 µM JC-1 and at 37 °C for 30 min. At the end of incubation, cells were washed to remove the staining medium. Fluorescence images were collected by using an inverted fluorescence microscope (DP72/CKX41, Olympus), and results were represented by the ratio of average red/green fluorescence intensity. Image Pro Plus 6.0 (Media Cybernetics, Rockville, MD, USA) software was used to measure red and green fluorescence and five random non-adjacent fields in each group were used for statistical analysis.

### 2.8. SA-β-Galactosidase Staining

Cells were rinsed with 1-mM MgCl_2_ in PBS (pH 6.0) and then stained for SA-β-galactosidase using a senescence β-galactosidase staining kit (Cell Signaling Technology, Danvers, MA, USA) according to the manufacturer’s instructions. After that, cells were washed twice with PBS and images were taken using a microscope (BX53, Olympus). SA-β-galactosidase positive cells were quantified by counting five random non-adjacent fields in each group.

### 2.9. Experimental Animals

12-week-old male Wistar rats were kept on a 12:12-h light/dark cycle with light from 7 a.m. to 7 p.m. and housed individually with free access to food and water. Rats were randomly divided into four groups, with six rats for each group. All animal experiments were approved by the Institutional Animal Care and Use Committee of Chung Shan Medical University, Taiwan (CSMU No. 2086) and performed in accordance with the guidelines and regulations of the Institutional Animal Care and Use Committee. In Nanog overexpression studies, the human wild-type Nanog coding sequence was amplified and cloned into the recombinant adeno-associated viral serotype 2 (rAAV2) vector. Intracerebral injection protocol and vector preparation were in reference to our previous studies [18]. For stereotaxic intracerebroventricular (i.c.v.) injections, rats were anesthetized with Zoletil (10 mg/kg, i.p.; Vibac Laboratories, Carros, France) and mounted in a stereotaxic frame (Stoelting Co., Ltd., Chicago, IL, USA). The rAAV-Nanog vector solution of 10 μL was injected into the left lateral cerebral ventricle with the following coordinates, 0.8 mm posterior to the bregma, 1.5 mm lateral from the midline and 3.8 mm ventral from the skull. Aβ_1–42_ solution of 5 µL (2 µg/µL, 10 µg each side) was injected into each side of hippocampus CA2 by using the following stereotaxic coordinates: 2.8 mm posterior to the bregma, 2.6 mm left/right to the midline and 3.0 mm ventral to the bregma. All the injections were performed within 5 min and following the needle remained in the target location for 10 min to avoid reflux along the needle tract. After 6 weeks of intracerebral injection, behavioral testing was performed. After that the animals were sacrificed. Brains were dissected and collected and then homogenized and fixed immediately.

### 2.10. T-Maze Test and Object Recognition Behavior Tests

Working memory was assessed using a T-maze test. The rats learned to find food rewards in the T-maze using their working memory, and the percentage of correct responses and time latency were recorded. After 6 weeks of injection, rats were subjected to behavioral tests performed as per our previous studies [19]. Each rat underwent two training sessions (on days 1 and 2) and one test session (on day 3); each training session consisted of nine trials. For object recognition tests, each rat was subjected to three exposure sessions at 24 h intervals (during days 4–6). Four different objects that were novel to the rats were presented before the experiment. Each rat was allowed to explore the objects in the open box for 5 min on three consecutive days. After 5 min of the last exposure session, one of the old objects was replaced by a novel one, and the animal was returned to the open box for a 5-min test session. Times spent exploring objects during exposure and test sessions were recorded. All data were collected by a SMART video tracking system (SMART v3.0, Panlab SL, Barcelona, Spain).

### 2.11. Statistical Analysis

All data are presented as means ± standard error of the mean (±SEM). Unpaired Student’s *t-*test performed correlations between two groups. For multiple comparisons, data were analyzed by one-way ANOVA with Dunnett’s post hoc multiple comparisons. F, DFn, DFd and p indicated the value of the F-test, the degrees of freedom of the numerator and denominator and the significance, respectively. Each group was compared with the indicated group. Data were analyzed by SPSS v25.0 statistical software (SPSS, Inc., Chicago, IL, USA). Differences were considered statistically significant while using * attached to indicate *p* < 0.05 and ** to indicate *p* < 0.01.

## 3. Results

### 3.1. Overexpression of Nanog Significantly Reduced Aβ-Mediated Cytotoxicity

To evaluate whether Nanog plays a role in enhancing neuroprotective effects against Aβ toxicity, we transiently transfected SK-N-MC neuronal cells with Nanog overexpression vector. Figure 1a confirms the results of qPCR that Nanog-transfected cells expressed markedly a higher Nanog mRNA level compared to mock-transfected cells (*p* < 0.01), indicating a successful overexpression of Nanog in SK-N-MC cells. western blot analysis also confirmed a significant induction of Nanog protein expression in transfected cells (Figure 1b). We next investigated whether overexpression of Nanog exerts protective effects against Aβ-induced neuronal death. As shown in Figure 1c by phase-contrast microscopy, treated with Aβ for 24 h caused marked cell death, whereas the overexpression of Nanog seemed to promote cell survival. Accordingly, MTT assays showed that treatment of cells with Aβ reduces cell viability in mock-transfected cells. Conversely, this cytotoxicity was significantly attenuated by Nanog overexpression (F(3, 12) = 14.07, *p* < 0.01, Figure 1d). To further determine which mode of cell death is induced by Aβ, results of DAPI staining showed that incubation of cells with Aβ appears to increase apoptotic nuclei fragmentation (F(3, 16) = 22.51, *p* < 0.01, Figure 1e). Similarly, alternative approaches by western blotting demonstrated that Aβ markedly increased cleavage of caspase 3 and poly (ADP-ribose) polymerase (PARP), two typical markers of apoptosis as shown in Figure 1f. However, these Aβ-induced apoptotic changes were effectively suppressed by overexpression of Nanog, suggesting Nanog overexpression significantly reduced Aβ-mediated cytotoxicity.

### 3.2. The Protection from Aβ-Induced Apoptosis by Nanog is Dependent on Insulin Signaling in SK-N-MC Neuronal Cells

Our previous study has shown that Aβ toxicity is involved in the blockade of insulin signaling in neuronal cells [14]. To determine whether overexpression of Nanog prevents Aβ-impaired neuronal insulin signaling, we performed western blotting to detect the level of insulin receptor substrate 1 (IRS-1) phosphorylation at Ser^307^, a typical marker linked to the severity of insulin resistance. As shown in Figure 2a, the serine-phosphorylated IRS-1 increased in cells with 2.5 µM Aβ treatment for 24 h, suggesting neuronal insulin signaling transduction is impaired by Aβ. Conversely, overexpression of Nanog caused a markedly decreased expression of serine-phosphorylated IRS-1. Similar results were also noted for tyrosine phosphorylation of IRS-1 that overexpression of Nanog dramatically enhances it in Aβ-treated group. Since the marker of insulin resistance was observed to be changed, in the next step we investigated if Nanog would affect downstream Akt activity. As shown in Figure 2b, Akt Ser^473^ phosphorylation was greatly diminished by Aβ. However, the reduced Akt Ser^473^ phosphorylation could be restored by overexpression of Nanog, suggesting Nanog can reverse Aβ-induced neuronal insulin signaling blockade. It is known that the phosphorylation at the residue Ser^9^ of glycogen synthase kinase 3β (GSK3β) by Akt can inhibit its kinase activity and the inhibition of GSK3β reduces tau hyperphosphorylation [20]. To elucidate the protective effect of Nanog against Aβ-induced cell death, the Ser^9^ phosphorylation levels of GSK3β were also evaluated. The results showed that the Ser^9^ phosphorylation of GSK3β was markedly suppressed by Aβ, indicating Aβ-inhibited IRS-1/Akt pathway results in GSK3β activation. However, the Aβ-blocked Ser^9^ GSK3β phosphorylation can be reversed by Nanog overexpression. This Nanog-mediated protection was also confirmed by inhibiting tau Thr^231^ phosphorylation—a crucial pathologic hallmark of AD (Figure 2c). In order to further investigate the role of IRS-1/Akt signaling in Nanog-mediated neuroprotection, the PI3-kinase inhibitor LY294002 was used as a negative control. As demonstrated in Figure 2d, LY294002 significantly blocked Nanog-prevented cell death by Aβ. MTT viability assays also displayed similar results that LY294002 significantly reduces the protective effects of Nanog (F(2, 9) = 13.49, *p* < 0.01, Figure 2e). Accordingly, the results of DAPI staining and western blots demonstrated that LY294002 downregulates the anti-apoptotic efficacy of Nanog (Figure 2f,g). Taken together, these results suggest that overexpression of Nanog effectively represses Aβ-induced tau phosphorylation and apoptosis by returning neuronal insulin signaling.

### 3.3. Aβ-Induced Superoxide Accumulation and Mitochondrial Dysfunction are Attenuated by Nanog Overexpression

Upon the basis of the above findings, we suggested that the protective effect of Nanog against Aβ is likely related to the restoration of insulin signaling blockade in neuronal cells. Therefore, we investigated whether Nanog protects cells from Aβ-induced oxidative stress by using the superoxide indicator dihydroethidium (DHE). As shown in Figure 3a, treatment with Aβ for 16 h caused a marked increase of superoxide accumulation and this increase was counteracted by overexpression of Nanog. It is known that the cellular redox state is mainly regulated by nuclear factor erythroid 2–related factor 2 (Nrf2), which can stimulate the expression levels of endogenous antioxidant enzymes such as superoxide dismutase 1 (SOD1) [21]. As expected, western blot results demonstrated that overexpression of Nanog effectively restores Aβ-induced reduction of Nrf2 and SOD1 levels (Figure 3b). However, co-treatment with LY294002 markedly attenuated Nanog-associated antioxidative effects including superoxide downregulation and Nrf2/SOD1 restoration, suggesting the involvement of insulin signaling in Nanog-mediated antioxidant activity. Because Aβ-induced mitochondrial dysfunction is known to cause oxidative damage in neurodegeneration [22], the role of Nanog in preventing Aβ-mediated impairment of mitochondrial membrane potential was also investigated. As shown in Figure 3c, Aβ treatment resulted in a strong increase in green fluorescence, indicating a great loss of mitochondrial membrane potential by Aβ. On the contrary, overexpression Nanog restored mitochondrial membrane potential significantly, suggesting Nanog preserves mitochondrial function against Aβ-induced mitochondrial dysfunction. Accordingly, this restoration was attenuated by co-treatment of LY294002, implying again that insulin signaling is required for the antioxidative actions of Nanog. Some evidence suggests that Aβ-induced oxidative stress mediates cellular senescence and contributes to AD pathogenesis [23]. As IGF-1 and BDNF exert a critical role of neurotrophins for survival of aged neurons that are degenerated in AD [24,25], we performed qPCR assays to determine differences in gene expressions with or without Nanog transfection. As shown in Figure 3d overexpression of Nanog upregulated both IGF-1 and BDNF mRNA expression, and this upregulation remained relatively high under Aβ treatment, suggesting that Nanog-transfected cells may increase neurotrophic factors production and improve neuronal insulin resistance caused by Aβ (F_IGF-1_(4, 15) = 6.05, *p* < 0.05; F_BDNF_(4, 15) = 13.49, *p* < 0.01). To elucidate whether Nanog can attenuate Aβ-induced cellular senescence, we performed SA-β-gal staining which is a common biomarker used in detecting senescent cells. The results of Figure 3e revealed a significant increase of SA-β-galactosidase positive cells in Aβ-treated cells, whereas Nanog effectively reduced the number of senescent cells (red arrows). Accordingly, Aβ (1.25 μM) caused a marked decrease in the expression of sirtuin-1 (Sirt1), a protein deacetylase that antagonizes cellular senescence (Figure 3f). This inhibition was effectively restored by overexpression of Nanog and abolished by combined treatment with LY294002. Collectively, our results indicate that restoration of mitochondrial function by Nanog may help reduce superoxide accumulation and protect neuronal cells from Aβ-induced senescence.

### 3.4. Overexpression of Nanog Protects Rat Brain against Aβ-Induced Cognitive Impairments

Finally, to investigate in vivo effects of Nanog-mediated neuroprotective potential against Aβ toxicity, we conducted an animal study based on intracerebral injection of Nanog expression vector into 12-week-old male Wistar rats. We stereotaxically delivered Nanog expression vector (rAAV-Nanog) into the brain ventricular space and Aβ_1–42_ solutions were also stereotaxically injected bilaterally into hippocampus CA2 to imitate neurotoxic actions of Aβ [18]. First, to evaluate the mRNA and protein levels of Nanog, hippocampus and cortex were homogenized and evaluated by using qPCR. As shown in Figure 4b, qPCR analysis revealed that the injection of Nanog expressing vector in rats significantly increases Nanog mRNA levels (~10-fold) in hippocampus (*p* < 0.01) and cortex (*p* < 0.05). Next, we examined whether exposure of rat hippocampal tissues to Aβ contributes to insulin signaling blockage. As shown in Figure 4c, western blot analysis revealed that exposure of hippocampal tissues to Aβ significantly induces IRS-1 Ser^307^ and suppresses Akt Ser^473^ phosphorylation, indicating that Aβ can lead to neuronal insulin resistance. However, overexpression of Nanog resulted in effective restoration of this neuronal insulin resistance. To further assess cognitive flexibility, behavioral tests were also performed at 6 weeks after surgery for working and recognition memory by using T-maze and object recognition tests, respectively. As shown in Figure 4d, T-maze test results demonstrated that the Aβ-only group shows a significantly low percentage of correct response, indicating Aβ impairs spatial learning (F(3, 20) = 3.45, *p* < 0.05). Conversely, a significant improvement was noted in the Nanog-overexpressed group. Latency to T-maze arm choice also exhibited similar results that the Nanog group features a shorter latency traveled before reaching the target of T-maze tests (F(3, 20) = 18.85, *p* < 0.01). These observations were further confirmed with findings from experiments of object recognition test (Figure 4e), wherein the results showed that rats in the Aβ-injected group spend significantly lower percentage of the time in exploring the novel object, indicating object recognition is damaged (F(3, 20) = 15.21, *p* < 0.01). However, this recognition impairment was effective restored in the Nanog-overexpressed group, suggesting that upregulation of Nanog in the brain may confer neuroprotection against Aβ-induced cognitive deficits.

## 4. Discussion

Downregulation of insulin signaling in the brain was considered as an important feature in the pathogenesis of AD. In particular, defective neuronal insulin signaling is strongly linked to Aβ-induced neurotoxicity [26]. In the brain, it is known that the insulin receptor is predominantly expressed in some specific regions including hippocampus, hypothalamus and cerebral cortex [27]. Notably, the principal role of insulin signaling in peripheral tissues is regulating the transport of glucose into the cell. However, researches have revealed that brain glucose uptake is not very greatly affected by insulin, suggesting insulin signaling may display some effects in the brain other than stimulation of glucose uptake [28]. In fact, numerous studies have demonstrated that brain insulin signaling is indeed involved in a broad spectrum of physiological functions. Especially in the hypothalamus, it is important for regulation of synaptic plasticity and long-term potentiation, two pathologic events occurring in AD progression [29]. As synaptic strength in the hippocampus is thought to be essential for the formation of spatial learning and memory, this indicates the dysfunction of neuronal insulin signaling may play a critical role in hippocampal synaptic plasticity impairment and cognitive decline. In contrast, restoration of insulin signaling can protect neurons from Aβ-induced neurotoxicity. With that, our results demonstrated that Aβ increases serine phosphorylation of IRS-1 and results in a concomitant reduction in Akt phosphorylation, consequently inhibiting insulin signaling in neuronal cells. On the contrary, overexpression of Nanog effectively attenuated Aβ-induced insulin signaling blockade and, thereby, improved the cell viability by downregulating oxidative stress and apoptosis. Interestingly, GSK3β is known as a major kinase to induce abnormal tau hyperphosphorylation, which is another contributor to induce neuronal cell death in AD. Because Ser^9^ phosphorylation is involved in the insulin signaling-mediated inhibition of GSK3β in neurons, Aβ-induced neuronal insulin resistance is now recognized as a significant feature in the pathogenesis of AD [30]. Accordingly, our results also demonstrated overexpression of Nanog effectively suppresses Aβ-induced tau phosphorylation and apoptosis by returning impaired neuronal insulin signaling. This further confirms that by increasing brain Nanog levels, insulin resistance and related cell death caused by Aβ can be reduced.

In addition, we also found when cells were incubated with Aβ, intracellular levels of Nanog was not inhibited by Aβ. This indicates the downregulation of Nanog may not play a central role of the process of neurodegeneration in AD brains. Actually, the aging process in the brain increases the senescence and loss of neurons. Given that aging neuron may be a consequence with impaired insulin signaling by Aβ, maintaining an efficient insulin sensitive in neurons can be a potential strategy for slowing neurodegeneration such as that seen in AD. As for how Nanog exerts its neuroprotection against Aβ toxicity, we previously showed that Nanog maintains the self-renewal of neuronal cells through the insulin signaling pathway [14]. Some other studies also demonstrated that the transplantation of genetically modified neuronal cells to overexpress neurotrophic factors can improve synaptic plasticity and restore memory impairment in AD mice [31]. Since stem cells have been identified as an important source of neurotrophic factors secretion, it is believed that maintaining the stemness of damaged neuronal cells may help the treatment of AD [32]. In addition, a remarkable decline of hippocampal neurogenesis during the process of aging is observed in vivo, suggesting that the increase of stemness may promote neuronal precursor cells to differentiate into functional neurons [33]. Evidence has also demonstrated that insulin signaling is necessary for maintaining stem cell self-renewal and pluripotency [34]. Therefore, methods for increasing stem cell activity can be considered as a potential approach for replacing lost cells in degenerative diseases. Since our results showed that Nanog enhances a stem-like phenotype in human neuronal cells, the anti-Aβ mechanism through which this occurs is worth to be used to inhibit disease progression of AD. For example, the use of some dietary phytochemical compounds that can stimulate Nanog expression may be applied in the future to develop the way in relieving Aβ-induced neurotoxicity [35].

Cellular redox state is regulated by antioxidant signaling involves cellular enzymatic system to scavenge excessive ROS production. In our research, we found that the Nanog-associated ROS inhibition, especially inhibiting the superoxide from mitochondria seems largely composed of Nrf2-regulated enzymes such SOD1 during Aβ incubation. It is known that Nrf2 antioxidant signaling is particularly involved in the survival of hippocampal neural stem cells, and upregulation of Nrf2 in the aging brain can also mitigate neurogenic decline and improve cognitive abilities [36]. Since brain Aβ deposition increases with age, these findings support our observations that activation of Nrf2 antioxidant system by Nanog may lead to a novel way for attenuating Aβ-induced toxicity and senescence. In addition, co-activation of the Nrf2 and neurotrophic signaling pathway is shown to generate a synergistic effect in slowing AD progress [37]. This finding is consistent with our current results that both the Nrf2 antioxidant system and neurotrophic pathway activated by Nanog may confer synergistic benefits to attenuate Aβ neurotoxicity. Because genetic manipulation of specific genes was shown to enhance antiaging properties of stem cells against exogenous stressors, upregulation of these genes such as Nanog involved in antioxidant restoration could be a feasible method to protect neurons from oxidative stress and associated damage [38]. Overall, the novelty of the present study lies in the cellular and behavioral analysis of Nanog’s neuroprotection against Aβ, as well as increased Nanog may be capable of supporting the survival and slowing aging of senescent neuronal cells. In conclusion, our results suggested a potential role for Nanog overexpression in modulating the neurotoxic effects of Aβ, which implies that strategies to enhance insulin signaling by overexpression of Nanog could be used as an intervention for AD treatment.

## Figures and Tables

**Figure 1 cells-09-01339-f001:**
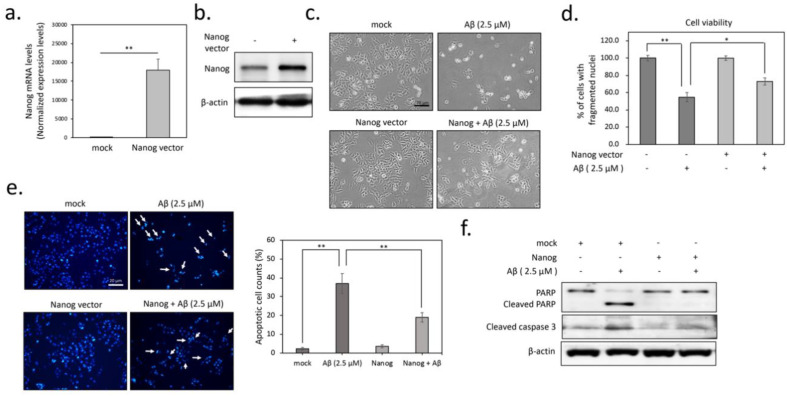
Overexpression of Nanog suppresses Aβ-induced apoptosis in human neuronal cells (SK-N-MC). Quantification of Nanog expression using real-time PCR (**a**) and western blot (**b**). At 24 h post-transfection, both mRNA and protein levels of Nanog were significantly upregulated, indicating the successful overexpression of Nanog; (**c**) treatment with 2.5 µM of Aβ_1–42_ for 24 h markedly induced morphologic changes and cell death and overexpression of Nanog prevented these effects by Aβ; (**d**) tetrazolium salt methyl-thiazol-tetrazolium (MTT) assays demonstrated that overexpression of Nanog significantly protected against Aβ-induced cytotoxicity; (**e**) overexpression of Nanog markedly reduced Aβ-induced nucleus fragmentation, determined using 4’,6-diamidino-2-phenylindole (DAPI) staining. The percentage of apoptotic cells was calculated from five random fields; (**f**) western blotting results revealed that overexpression of Nanog inhibited Aβ-induced caspase-3 and polymerase (PARP) cleavages, two hallmarks of apoptosis. Values were expressed as means ± SEM from at least three independent experiments. The significance of differences was determined through unpaired Student’s *t-*test or multiple comparisons with Dunnett’s post hoc test at * *p* < 0.05 and ** *p* < 0.01 compared with the indicated groups. Scale bar represents 20 μm.

**Figure 2 cells-09-01339-f002:**
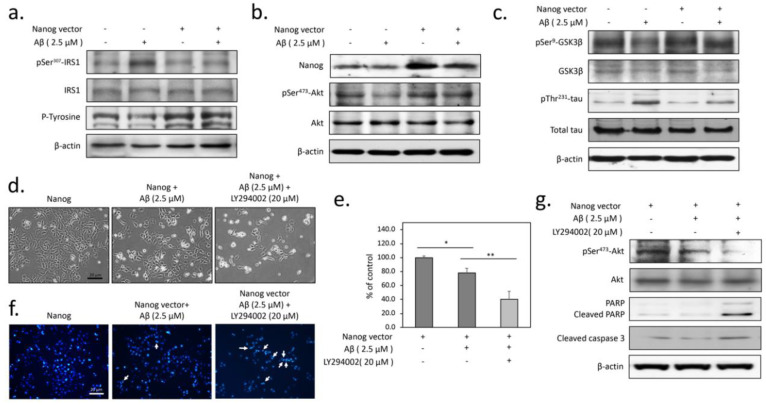
Overexpression of Nanog inhibits Aβ-induced tau phosphorylation and cytotoxicity by restoring impaired insulin signaling; (**a**) western blots showed that treatment with 2.5 µM of Aβ for 24 h induces a marked increase of insulin receptor substrate 1 (IRS-1) Ser^307^ phosphorylation, a major marker of insulin resistance. However, overexpression of Nanog greatly inhibited this phosphorylation; (**b**) western blot analysis of Ser^473^ phosphorylated Akt confirmed that overexpression of Nanog reverses Aβ-induced insulin signaling blockade; (**c**) western blot results indicated that overexpression of Nanog inhibits Aβ-induced tau phosphorylation at Thr^231^ and increases the phosphorylation of glycogen synthase kinase 3β (GSK3β) at Ser^9^; (**d**) Bright field images showed that Nanog-induced protection is abolished by the co-treatment with LY294002 (20 µM), a specific inhibitor of PI3-kinase; (**e**) MTT assays showed the LY294002 co-treatment inhibits the protective effect of Nanog; (**f**) LY294002 markedly reduced Nanog-induced anti-nucleus fragmentation effects determined by using DAPI staining. The arrows indicated apoptotic cells with nuclear fragmentation; (**g**) western blots showed that LY294002 markedly prevents the reduction of caspase-3 and PARP cleavages, suggesting the protection by Nanog is dependent on insulin signaling. Values were expressed as means ± SEM from at least three independent experiments. The significance of differences was determined through multiple comparisons with Dunnett’s post hoc test at * *p* < 0.05 and ** *p* < 0.01 compared with the indicated groups. Scale bar represents 20 μm.

**Figure 3 cells-09-01339-f003:**
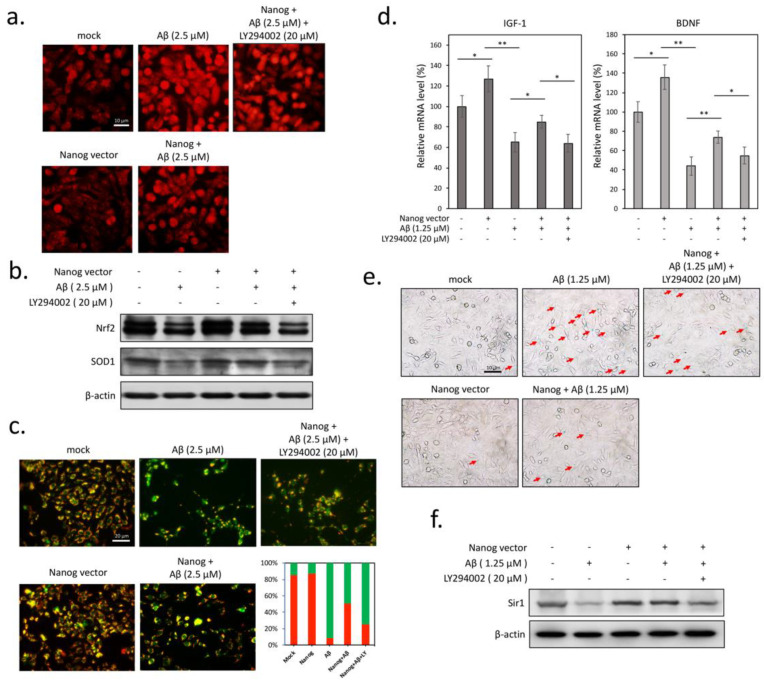
Nanog overexpression ameliorates Aβ-induced mitochondrial superoxide accumulation and cellular senescence. (**a**) Dihydroethidium (DHE) staining results showed that overexpression of Nanog reduces Aβ-induced intracellular superoxide accumulation; (**b**) levels of two antioxidant signaling-related proteins nuclear factor erythroid 2-related factor 2 (Nrf2) and superoxide dismutase 1 (SOD1) were analyzed by western blotting; (**c**) double fluorescence staining of mitochondrial membrane potential by JC-1 was used for detection of mitochondrial membrane potential. Green fluorescence indicated the decreased membrane potential in 2.5 µM of Aβ-treated SK-N-MC cells after 24 h. Red fluorescence indicated that overexpression of Nanog effectively preserves the mitochondrial membrane potential; (**d**) mRNA levels of neurotrophic factors insulin-like growth factor 1 (IGF-1) and brain-derived neurotrophic factor (BDNF) were measured by performing qPCR. mRNA levels of IGF-1 and BDNF were significantly increased in Aβ treated Nanog-overexpressed cells; (**e**) representative results of cytochemical detection of SA-β-galactosidase, a common biomarker used in detecting senescent cells. Senescent cells (red arrows) are blue stained under a bright-field microscope; (**f**) levels of sirtuin-1 (Sirt1) protein in Aβ-treated cells using western blotting. Values were expressed as means ± SEM from at least three independent experiments. The significance of differences was determined through multiple comparisons with Dunnett’s post hoc test at * *p* < 0.05 and ** *p* < 0.01 compared with the indicated groups. LY294002, a PI3K inhibitor for reducing insulin signaling.

**Figure 4 cells-09-01339-f004:**
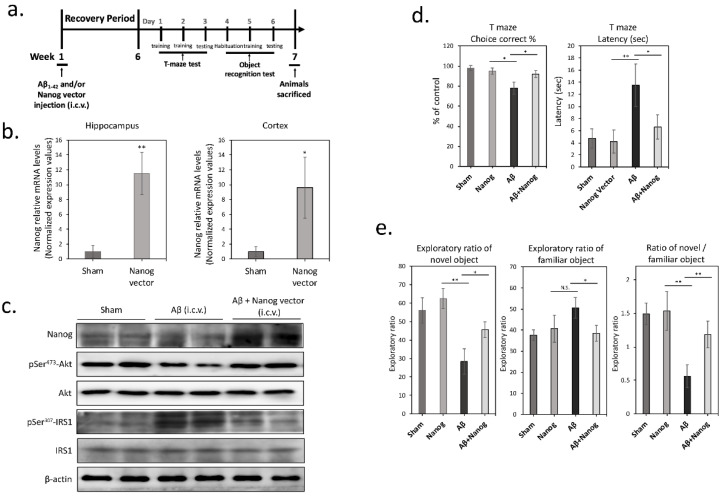
Upregulation of Nanog in rat brains improves working and recognition memory deficits induced by Aβ. (**a**) The experimental protocol of the behavioral tests; (**b**) six weeks after stereotaxical injection, mRNA levels of Nanog were measured by performing qPCR. Nanog mRNA levels in hippocampus and cortex were significantly increased in Nanog-overexpressed group; (**c**) western blotting revealed that overexpression of Nanog markedly inhibited IRS-1 Ser^307^ and restored Akt Ser^473^ phosphorylation in rat hippocampus; (**d**) behavioral testing was performed at 6 weeks after stereotaxical surgery. Results showed the percentage of correct responses to select the arm with the reward and time latency to finish each session for the T-maze test; (**e**) percentage of time spent exploring new or old objects in object recognition test. Upregulation of Nanog in the brain significantly increased the ratio of novel/familiar object exploring time compared to Aβ-injected group. Values were expressed as means ± SEM from at least three independent experiments. The significance of differences was determined through unpaired Student’s *t-*test or multiple comparisons with Dunnett’s post hoc test at * *p* < 0.05 and ** *p* < 0.01 compared with the indicated groups. N.S.: not significant.

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
