# Peer review of "The Pluripotency Factor Nanog Protects against Neuronal Amyloid β-Induced Toxicity and Oxidative Stress through Insulin Sensitivity Restoration"

_cells, 2020, doi:10.3390/cells9061339_

Round 1

Reviewer 1 Report

The manuscript by Chang and colleagues is focused on the Nanog and its relationship to amyloid beta-induced toxicity.

This is an interesting paper that is well written, the methodology is appropriately chosen and the conclusions are based on the well-presented results. The only insufficiency of this manuscript is a better determination of antibodies quality as antibodies usually suffer by cross-reactivity to other similar molecules. Has been this proved by the manufacturer/other studies?

Minor comments:

1) This referee suggest presenting experimental scheme for faster orientation in experimental design,

2) Line 163: lights on, lights off should be given,

3) line 196-197: the type of ANOVA should be given: one-way, two-way?

4) ANOVA results should be more specific, i.e. F, dFd etc. should be presented,

2) The manuscript should be checked again for syntax/grammatical errors/typos: e.g.: line 82-87 duplicate verb, line 440 F For research...

Author Response

Responses to the Reviewers' comments (cells-816784) R1

Comments to the Authors:

#Reviewer 1

The manuscript by Chang and colleagues is focused on the Nanog and its relationship to amyloid beta-induced toxicity.

This is an interesting paper that is well written, the methodology is appropriately chosen and the conclusions are based on the well-presented results. The only insufficiency of this manuscript is a better determination of antibodies quality as antibodies usually suffer by cross-reactivity to other similar molecules. Has been this proved by the manufacturer/other studies?

Answer: Thanks for reviewer’s important comments. We understand your concern about the role ofantibodies quality. In fact, most of the antibodies used in this study, including IRS-1, Akt, GSK3β, Nrf2, Sirt1, SOD, and apoptosis-related Abs etc., have been used in our recently published papers. (J. Cell. Physiol. 234:9733-9745, 2019; J. Cell. Mol. Med. 23:619-629, 2019; Int. J. Mol. Sci. 19:2505, 2018; CNS Neurosci. Ther. 24:47-57, 2018 etc.) Most antibodies were good for our results, and no serious cross-reaction was found. Some publications were also cited by antibody manufacturers. However, the Nanog antibody really caused us some trouble. We tried many manufacturers, including SantaCruz, Cell Signaling, and GeneTex, etc., the quality is not very stable until the problem was solved after we purchased Millipore’s product. In this regard, we have attached all the antibody Cat. No. used in this revised version, and hope that can be used as a reference for readers to carry out relevant experiments later.

Minor comments:

1) This referee suggest presenting experimental scheme for faster orientation in experimental design,

Answer: According to reviewer’s instruction, we have added an animal experimental scheme figure in Fig. 4a. We hope this could help readers better understand the flow chart of animal experiments.

2) Line 163: lights on, lights off should be given,

Answer: To provide a clear time to light on and light off the lights, the description has been modified to “12-week-old male Wistar rats were kept on a 12:12-h light/dark cycle with light from 7 a.m. to 7 p.m., …”. (page 4, line 166-167)

3) line 196-197: the type of ANOVA should be given: one-way, two-way?

4) ANOVA results should be more specific, i.e. F, dFd etc. should be presented,

Answer: We are sorry that the statistical methods is not clear enough. We have now amended the statistical description to “All data are presented as means ± standard error of the mean (±SEM). Unpaired Student’s t-test performed correlations between two groups. For multiple comparisons, data were analyzed by one-way ANOVA with Dunnett’s post hoc multiple comparisons. F, DFn, DFd and p indicated the value of the F-test, the degrees of freedom of the numerator and denominator and the significance, respectively. Each group was compared with the indicated group. Data were analyzed by SPSS v25.0 statistical software (SPSS, Inc., Chicago, IL, USA).” by reviewer ’s suggestion. In principle, if only two sets of data are to be compared, we take the method of unpaired Student ’s t-test (such as Fig. 1a and Fig. 4b). For comparison of more than two sets of data, we took the one-way ANOVA method, and conducted Dunnett ’s post hoc for groups with significant differences. In addition, we also provided F, DFn, DFd and p values in the text, and hope these corrections will make it easier for readers to evaluate our results.

2) The manuscript should be checked again for syntax/grammatical errors/typos: e.g.: line 82-87 duplicate verb, line 440 F For research...

Answer: Thanks for reviewer’s comments. All typographical and grammar errors have been carefully corrected (including what you have pointed out), and some vague or inadequate statements in text are also rewritten. We hope these modifications will make our results clearer and more reliable, and would like to accept any further suggestions raised by reviewers. Once again, thank you very much for your insightful comments of our manuscript.

Reviewer 2 Report

The manuscript describes the effect of pluripotency factor Nanog on the protection against neuronal amyloid β-induced toxicity and oxidative stress through insulin sensitivity restoration. The lead action in the work done and presented is exemplified by Nanog, with the intervention in signaling pathways associated with insulin activity in neuronal cells, sensitive apoptotic events in neurodegeneration (such as that encountered in Alzheimer’s disease), being a prominent feature. The work is done competently, yet a number of issues arise that need to be corrected and clarified prior to any commitment.

  1. Linguistic errors throughout the text appear to cause problems to the reader, when following the statements and ideas of the authors pertaining to the experimental data and their interpretation. The manuscript is in a dire need of a linguistic revision
  2.  In the Abstract, it is stated that “Amyloid β (Aβ) is best known as a major neurotoxic protein to cause Alzheimer's disease 18 (AD).”. Aβ is not a neurotoxin protein. It is a neurotoxic peptide originating in the Amyloid Precursor Protein. That statement should be corrected.
  3. In the same section, the statement “Nanog is recognized as a stem cell-related pluripotency factor, which enhances self-renewing capacities and helps to reduces the senescent phenotypes of aged neuronal cells.” should be corrected to read “Nanog is recognized as a stem cell-related pluripotency factor, which enhances self-renewing capacities and helps reduce the senescent phenotypes of aged neuronal cells.”.
  4. In the Materials and Methods section, a statement reads “Aβ1-42 was synthesized by LifeTein (Somerset, NJ, USA), and Aβ lyophilized peptides were prepared and resuspended in anhydrous dimethyl sulfoxide (DMSO) as our previously described [17].”. It is not clear which other peptides the authors refer to beyond the amyloid peptide. Clarification of the statement is required.
  5. In section “2.6 Detection of superoxide by dihydroethidium (DHE) staining” the statement “At the end of experiments, cells were rinsing with PBS solution and then incubated in fresh media containing 10 μM DHE solution for 30 min in the dark at room temperature.” should be corrected to read “At the end of experiments, cells were rinsed with PBS solution and then incubated in fresh media containing 10 μM DHE solution for 30 min in the dark at room temperature.”.
  6. In section 3.3, the chemical used to probe whether Nanog protects cells from Aβ-induced oxidative stress was the superoxide indicator dihydroethidium (DHE). The proper correction should be made.
  7. In section “3.4 Overexpression of Nanog protects rat brain against Aβ-induced cognitive impairment” the statement “As shown in Fig. 4B, western blot analysis revealed that exposure Aβ to hippocampal tissues significantly induces IRS-1 Ser307 and suppressed Akt Ser473 phosphorylation, indicating Aβ can lead to neuronal insulin resistance.” should be corrected to read “As shown in Fig. 4B, western blot analysis revealed that exposure of hippocampal tissues to Aβ significantly induces IRS-1 Ser307 and suppresses Akt Ser473 phosphorylation, indicating that Aβ can lead to neuronal insulin resistance.”.
  8. In the beginning of the Discussion, the statement “Downregulation of insulin signaling in the brain has been considered as important features in the pathogenesis of AD.” is not understood. Did the authors mean “Downregulation of insulin signaling in the brain has been considered as an important feature in the pathogenesis of AD.”? Clarity is required.
  9. In the same section, the paragraph “In addition, a pronounced decline in hippocampal neurogenesis of the aging brain is observed in vivo, suggesting that increasing the stemness may also have the ability to growth in the number of stem cells and replace lost cells by differentiating into functional neurons [33]. It is noteworthy the insulin signaling has been recognized as a fundamental pathway contributes to stem cell-like properties such as self-renewal and pluripotency [34].” is confusing and ill-understood. It should be re-written in a clear fashion.
    Based on the aforementioned grounds, the paper could be considered further provided that all of the above remarks are addressed appropriately.

Author Response

Responses to the Reviewers' comments (cells-816784) R1

Comments to the Authors:

#Reviewer 2

The manuscript describes the effect of pluripotency factor Nanog on the protection against neuronal amyloid β-induced toxicity and oxidative stress through insulin sensitivity restoration. The lead action in the work done and presented is exemplified by Nanog, with the intervention in signaling pathways associated with insulin activity in neuronal cells, sensitive apoptotic events in neurodegeneration (such as that encountered in Alzheimer’s disease), being a prominent feature. The work is done competently, yet a number of issues arise that need to be corrected and clarified prior to any commitment.

Linguistic errors throughout the text appear to cause problems to the reader, when following the statements and ideas of the authors pertaining to the experimental data and their interpretation. The manuscript is in a dire need of a linguistic revision

In the Abstract, it is stated that “Amyloid β (Aβ) is best known as a major neurotoxic protein to cause Alzheimer's disease 18 (AD).”. Aβ is not a neurotoxin protein. It is a neurotoxic peptide originating in the Amyloid Precursor Protein. That statement should be corrected.

Answer: Thank you very much for your kind and careful corrections. For the errors you pointed out above,we have modified it to “Amyloid β (Aβ) is a peptide fragment of the amyloid precursor protein that triggers the progression of Alzheimer's Disease (AD)”, according to the reviewer's suggestion (page 1, line 18-19).

In the same section, the statement “Nanog is recognized as a stem cell-related pluripotency factor, which enhances self-renewing capacities and helps to reduces the senescent phenotypes of aged neuronal cells.” should be corrected to read “Nanog is recognized as a stem cell-related pluripotency factor, which enhances self-renewing capacities and helps reduce the senescent phenotypes of aged neuronal cells.”.

Answer: We have corrected it according to the reviewer's suggestion. (page 1, line 23-25)

In the Materials and Methods section, a statement reads “Aβ1-42 was synthesized by LifeTein (Somerset, NJ, USA), and Aβ lyophilized peptides were prepared and resuspended in anhydrous dimethyl sulfoxide (DMSO) as our previously described [17].”. It is not clear which other peptides the authors refer to beyond the amyloid peptide. Clarification of the statement is required.

Answer: According to reviewer ’s comment, we have changed the description to “…soluble Aβ oligomers were prepared according to our previously described [17]. Briefly, Aβ was first dissolved in hexafluoroisopropanol (HFIP), distributed in aliquots, dried and stored at −80°C. The day before the experiment, Aβ was dissolved in sterile dimethylsulfoxide (DMSO), then diluted with PBS (pH 7.4) at a final concentration of 100 μM, vortexed and incubated at 4°C for 24 h.” We hope this amendment could help readers to better understand the preparation protocol of Aβ solution. (page 2, line 91-95)

In section “2.6 Detection of superoxide by dihydroethidium (DHE) staining” the statement “At the end of experiments, cells were rinsing with PBS solution and then incubated in fresh media containing 10 μM DHE solution for 30 min in the dark at room temperature.” should be corrected to read “At the end of experiments, cells were rinsed with PBS solution and then incubated in fresh media containing 10 μM DHE solution for 30 min in the dark at room temperature.”.

Answer: We have modified it according to the reviewer's suggestion. (page 3, line 144-145)

In section 3.3, the chemical used to probe whether Nanog protects cells from Aβ-induced oxidative stress was the superoxide indicator dihydroethidium (DHE).

In section “3.4 Overexpression of Nanog protects rat brain against Aβ-induced cognitive impairment” the statement “As shown in Fig. 4B, western blot analysis revealed that exposure Aβ to hippocampal tissues significantly induces IRS-1 Ser307 and suppressed Akt Ser473 phosphorylation, indicating Aβ can lead to neuronal insulin resistance.” should be corrected to read “As shown in Fig. 4B, western blot analysis revealed that exposure of hippocampal tissues to Aβ significantly induces IRS-1 Ser307 and suppresses Akt Ser473 phosphorylation, indicating that Aβ can lead to neuronal insulin resistance.”.

Answer: We have corrected it by reviewer's comment. (page 7, line 292; page 9, line 352-354)

In the beginning of the Discussion, the statement “Downregulation of insulin signaling in the brain has been considered as important features in the pathogenesis of AD.” is not understood. Did the authors mean “Downregulation of insulin signaling in the brain has been considered as an important feature in the pathogenesis of AD.”? Clarity is required.

Answer:

We have modified it by reviewer's suggestion. (page 10, line 385-386)

In the same section, the paragraph “In addition, a pronounced decline in hippocampal neurogenesis of the aging brain is observed in vivo, suggesting that increasing the stemness may also have the ability to growth in the number of stem cells and replace lost cells by differentiating into functional neurons [33]. It is noteworthy the insulin signaling has been recognized as a fundamental pathway contributes to stem cell-like properties such as self-renewal and pluripotency [34].” is confusing and ill-understood. It should be re-written in a clear fashion.

Based on the aforementioned grounds, the paper could be considered further provided that all of the above remarks are addressed appropriately.

Answer: Thanks for reviewer’s comment. We have rewritten this text as “…In addition, a remarkable decline of hippocampal neurogenesis during the process of aging is observed in vivo, suggesting that the increase of stemness may promote neuronal precursor cells to differentiate into functional neurons [33]. Evidence has also demonstrated that insulin signaling is necessary for maintaining stem cell self-renewal and pluripotency [34]. Therefore, methods for increasing stem cell activity can be considered as a potential approach for replacing lost cells in degenerative diseases.” (page 11, line 422-427) We hope this correction can reduce the confusion that may cause to readers. Once again, thank you very much for your valuable comments, and we would like to accept any further suggestion raised by reviewers.
